# Is Visual Pedagogy Effective in Improving Cooperation towards Oral Hygiene and Dental Care in Children with Autism Spectrum Disorder? A Systematic Review and Meta-Analysis

**DOI:** 10.3390/ijerph18020789

**Published:** 2021-01-18

**Authors:** Araxi Balian, Silvia Cirio, Claudia Salerno, Thomas Gerhard Wolf, Guglielmo Campus, Maria Grazia Cagetti

**Affiliations:** 1Department of Biomedical, Surgical and Dental Science, University of Milan, Via Beldiletto 1, 20142 Milan, Italy; araxibalian@gmail.com (A.B.); silviacirio@alice.it (S.C.); claudia.salerno@ymail.com (C.S.); 2Department of Restorative, Preventive and Pediatric Dentistry, University of Bern, Freiburgstrasse 7, 3012 Bern, Switzerland; thomas.wolf@zmk.unibe.ch (T.G.W.); guglielmo.campus@zmk.unibe.ch (G.C.); 3Department of Periodontology and Operative Dentistry, University Medical Center of the Johannes Gutenberg-University Mainz, 55131 Mainz, Germany; 4Department of Surgery, Microsurgery and Medicine Sciences, School of Dentistry, University of Sassari, Viale San Pietro 3/c, 07100 Sassari, Italy; 5School of Dentistry, Sechenov University, 119991 Moscow, Russia

**Keywords:** visual pedagogy, autism spectrum disorders, dental setting, oral hygiene, dental care

## Abstract

Visual pedagogy has emerged as a new approach in improving dental care in children with autism spectrum disorders (ASDs). This paper aimed to evaluate and assess the scientific evidence on the use of visual pedagogy in improving oral hygiene skills and cooperation during dental care in children with ASDs. The review protocol was registered on the PROSPERO Register (CRD42020183030). Prospective clinical studies, randomized trials, interruptive case series, before and after comparison studies, and cross-sectional studies following the PRISMA guideline were searched in PubMed, Embase, Scopus, and Google Scholar using ad hoc prepared search strings. The search identified 379 papers, of which 342 were excluded after title and abstract evaluation, and 37 full-text papers were analyzed. An additional four papers were added after consulting reference lists. Eighteen papers were disregarded; 23 were finally included, and their potential bias was assessed using ROB-2 and ROBINS-I tools. The wide heterogenicity of the studies included does not allow for conclusive evidence on the effectiveness of visual pedagogy in oral hygiene skills and dental care. Nevertheless, a significant and unilateral tendency of the overall outcomes was found, suggesting that visual pedagogy supports ASD children in improving both oral hygiene skills and cooperation during dental care.

## 1. Introduction

Autism spectrum disorders (ASDs) have progressively acquired more and more dignity and importance in the world health panorama, even in the dental field. Children with ASD are greatly challenged when facing new experiences, and the dental environment is of particular concern due to the presence of several noises, smells, and visual stimuli that might exacerbate fear and anxiety [1,2,3]. Concerns about dental care may lead parents to avoid regular dental examinations [4]. A good level of oral hygiene is quite difficult to maintain, since they often refuse brushing and flossing [5,6], increasing the risk of dental caries and gingivitis compared to children not affected by ASD. In addition to poor oral hygiene, a high frequency of sugary food and beverage consumption is frequently reported [6,7].

Behavioral management techniques derived from pediatric dentistry practice (desensitization, positive-negative reinforcement, tell-show-do) have been used to improve the ability of children with ASDs to receive dental treatment and oral healthcare [8,9]. This approach is the first attempt in a series of approaches to overcome undesirable behaviors during oral examinations and dental procedures. Behavioral management can be effective for some, but not for every patient. Many children with ASDs still require advanced behavioral guidance techniques, such as protective stabilization, oral sedation, and general anesthesia, to provide dental care [10]. Behavioral approaches are the most common treatment approaches for children with ASDs, and interventions often include the use of visual pedagogy. It is defined as the ability to recognize and understand ideas conveyed through visible actions or images [11], and it can be used to enable and/or increase specific skills of children [10]. The method involves the use of pictures/imagines either printed on paper or administered though digital tools, such as computers, smartphones, and tablets; such feasible interactive aids are becoming more and more utilized with special needs children. Among the different visual tools available, the Picture Exchange Communication System (PECS) is a frequently used augmentative communication system, in which picture cards are used to teach functional communication to non-verbal or limited speech children [12]. Visual pedagogy protocols foresee the use of sketches and/or videos to repetitively teach children how to perform tooth brushing and which steps they will encounter during oral examinations and preventive and/or restorative treatments. The core of visual pedagogy is that children with ASDs become familiar with the storytelling that they will remember when in the dental office. A high number of studies have already been carried out on this approach, proving this to be effective in reducing anxiety and increasing compliance [4,5,13].

The purpose of this paper is the evaluation and grading of the scientific evidence of the existing literature on the use of visual pedagogy as a strategy for improving oral hygiene skills in children with ASDs. The effect of visual pedagogy on children’s cooperation during dental care was also assessed. A systematic review and meta-analysis were designed and carried out for this purpose.

## 2. Materials and Methods

This review follows the Preferred Reporting Items for Systematic Reviews and Meta-Analyses (PRISMA) guideline [13]. The review protocol was registered on the International Prospective Register of Systematic Reviews (PROSPERO) with registration number CRD42020183030. The question was structured and focused according to the PICO format (Population, Intervention, Comparison, and Outcome):
Population: Children with autism spectrum disorders;Intervention: Effect of visual pedagogy;Comparison: Visual pedagogy vs. no treatment or outcomes measured before and after visual pedagogy administration;Outcome: Oral hygiene skills (primary outcome) and/or cooperation during dental care (secondary outcome).

### 2.1. Eligibility Criteria

The inclusion criteria were:
Type of study: prospective clinical studies, randomized trials, interruptive case series, before and after comparison studies, cross-sectional studies;Publication languages: papers published in English, Italian, and French;Time of publication: no time restriction applied, last accessed on 23 July 2020;Type of tool used: PECS, images on paper, such as dental books, picture cards, drawings, and printed photos, or on digital supports, such as tablets, dental apps, and/or videos;Primary outcome: clinical indices of oral hygiene skills, such as the plaque index (PI) and the gingival index (GI). Tooth brushing performance was also considered.Secondary outcome: indices of patient’s cooperation level during dental procedures, such as the Frankl Behavior Score and the Likert Anxiety Scale, and/or the number of steps/procedures completed and time spent, measured by a dentist or a dental hygienist and/or a psychologist/educator.

### 2.2. Information Sources and Search Strategy

Four electronic databases were searched from the inception of each database until 23 July 2020, and Medline via PubMed, Embase via Ovid, Scopus, and Google Scholar were screened. The search strategy included a search string for each electronic database selected. For Medline via Pubmed, the string used was: (audiovisual aids[mh] or “visual pedagogy”[tiab] or “social story”[tiab] or “audio modeling”[tiab] or “visual modeling”[tiab] or “video modeling”[tiab] or pecs[tiab] or tablet[tiab] or ipad[tiab] or “audiovisual distraction”[tiab] or “visual support”[tiab] or “patient education as topic”[mh] or “behavior therapy”[mh] or desensitization [mh] or “sensory”[tiab] or “preparatory aid”[tiab] or “pictures” [tiab] or “dental book”) and (autism spectrum disorder [mh] or autism or asd or “special need”) and (dent * or “oral health” or “dental care” or “oral hygiene” OR “oral” OR “dental”); for Embase via Ovid (’audiovisual aid’/exp/mj OR ‘audiovisual aid’ OR ‘visual system’; [tiab] OR ‘pedagogics’ OR ‘social story’ OR ‘audiovisual equipment’ OR ‘tablet computer’ OR ‘patient education’ OR ‘behavior therapy’ OR ‘visual aid’ OR ‘picture exchange communication system’) AND (‘autism’) AND (‘oral health care’ OR ‘oral health status’ OR ‘dentistry’ OR ‘mouth hygiene’ OR ‘tooth brushing’); for Scopus: INDEXTERMS (“audiovisual aids”) OR TITLE-ABS (“visual pedagogy”) OR TITLE-ABS (“social story”) OR TITLE-ABS (“audio modeling”) OR TITLE-ABS (“visual modeling”) OR TITLE-ABS (“video modeling”) OR TITLE-ABS (pecs) OR TITLE-ABS (tablet) OR TITLE-ABS (ipad) OR TITLE-ABS (“audiovisual distraction”) OR TITLE-ABS (“visual support”) OR INDEXTERMS (“patient education as topic”) OR INDEXTERMS (“behavior therapy”) OR INDEXTERMS (desensitization) OR TITLE-ABS (sensory) OR TITLE-ABS (“preparatory aid”) OR TITLE-ABS (pictures) OR TITLE-ABS (“dental book”) AND INDEXTERMS (“autism spectrum disorder”) OR autism OR asd OR “special need” AND INDEXTERMS (dental) OR “oral health” OR “dental care” OR “oral hygiene” OR oral OR dental; finally, for Google Scholar the string was as follows: autism OR ASD OR “autistic spectrum disorder” OR “special child” dental OR “oral hygiene” OR “tooth brushing” OR “Oral Health”. Cross-referencing was also performed using the references lists of full-text articles. Grey literature was also retrieved via opengrey.eu (http://www.opengrey.eu).

### 2.3. Study Selection

The output of the reference searches was uploaded into Excel software 16.16 (Microsoft, Redmond, WA, USA), and duplicates were excluded after comparing the results from the different research strategies. Four authors (A.B., S.C., C.S., and T.G.W.) independently examined all of the abstracts; papers meeting the inclusion criteria were obtained in the full-text format. The authors independently assessed the papers to establish whether each paper should or should not be included in the systematic review. Disagreements were resolved through discussion and/or by full-text analysis in doubtful cases. Where resolution was not possible, another author was consulted (M.G.C.).

### 2.4. Data Collection, Summary Measures, and Synthesis of Results

Data collection and synthesis were independently carried out by four authors (T.G.W., A.B., C.S., and S.C.) using an ad hoc designed data extraction form (Appendix A extraction form), without masking the name of the journal, title, or authors. Studies selected were divided into two groups according to their primary outcome. In the first group, articles that investigated the effectiveness of visual pedagogy in improving oral hygiene skills in children with ASDs were included [14,15,16,17,18,19,20,21,22]. In the second group, articles that investigated the effectiveness of visual pedagogy in improving the patient’s cooperation during dental care were included [4,23,24,25,26,27,28,29,30,31,32,33,34,35]. To facilitate the synthesis, the results were summarized in tables. For each paper, these data were searched and recorded when available: (a) source, publication year, location, and study duration; (b) details/characteristics of the participants; (c) level of disability/verbal fluency; (d) type of tool used and visual pedagogy protocol and adjunctive tool when used.

### 2.5. Quality Assessment and Scientific Evidence

The risk of bias assessment was performed by three authors (M.G.C., A.B., and S.C.), and the Cochrane Risk of Bias tools for randomized and non-randomized studies were used for methodological quality evaluation. A per-protocol analysis was conducted with the aim of assessing the effect of starting and adhering to the intervention. The Cochrane collaboration’s ROB-2 tool was used to assess the risk of bias for randomized studies [36]. The Excel (Microsoft Corporation, Washington, U.S.) tool for ROB-2 was used to input answers given to signaling questions, and then an algorithm estimated the overall risk of the bias according to the results for each domain as: low risk, some concerns, or high risk. The risk of bias plots were drawn using the Cochrane robvis web app [37]. The Cochrane collaboration’s ROBINS-I tool was used to assess the risk of bias for non-randomized studies of intervention (NRSI) [38]. Authors answered signaling questions in each domain, and then estimated the overall risk of the bias according to the results for each domain as: low, moderate, serious, or critical.

A list of criteria was agreed upon by three authors (M.G.C., A.B., and S.C.) to be followed in bias assessment for both RCT and NRSI. The standardization of the research protocol was considered challenging, and it was not considered in a strict manner due to the need to frequently adopt individual, case-based strategies in approaching patients with ASDs [25]. A list of confounding domains and co-interventions was agreed upon, and they were identified as: type and severity of ASD; age; previous use of visual pedagogy; and the presence of a control group. Bias related to deviation from treatment protocol was rated as low if visual pedagogy was administered by health personnel, as moderate if it was administered at home and compliance was verified, and as serious/critical if visual pedagogy was provided at home and cooperation was not verified. The presence of drop-outs was of particular interest both in randomized and non-randomized studies, since no intention-to-treat analysis (ITT) was performed in any study. Drop-outs were judged as follows: drop-outs less than 10%, low risk; drop-outs of 10–20%, moderate risk; drop-outs of 20–30%, serious risk; drop-outs more than 30%, critical risk. Blinding is more often difficult in such studies, and was rated as follows: double blinding, low risk; single blinding, moderate risk; no blinding, serious risk. The risk of bias assessment was evaluated independently by three reviewers (A.B., S.C., and T.G.W) and then discussed together with a third reviewer (M.G.C.) in order to resolve disagreements and provide the overall final judgment for each study.

### 2.6. Statistical Analysis

STATA16 Software (Statacorp, College Station, TX, USA) was used for the meta-analysis of the data. The mean difference (MD) and odds ratio (OR) were chosen to calculate the effect size. The analysis was computed on the different visual tools used. A meta-analysis was performed if two or more studies compared the effect of visual pedagogy using comparable outcomes (G.C.). The I2 statistic was calculated to describe the percentage of variation across studies due to heterogeneity rather than chance [39]. The heterogeneity was categorized as follows: <30%, not significant; 30–50%, moderate; 51–75%, substantial, and 76–100%, considerable. Whether homogeneity was obtained or not, the random effects model (REM) with 95% confidence intervals was chosen as the meta-analysis model.

## 3. Results

### 3.1. Study Selection

The search identified 478 papers; 379 were selected after removing duplicates, then 342 papers were excluded after a title and abstract evaluation (Appendix A, List of excluded papers after the first evaluation). Thirty-seven papers were obtained in their full-text format, and an additional four papers were added after consulting the references lists (Figure 1).

Therefore, forty-one papers were assessed; eighteen papers were discarded (Appendix A). Twenty-three studies were finally included in this systematic review: nine studies concerned tooth brushing and oral hygiene skills in children with ASDs and 14 studies concerned their cooperation during dental procedures (Figure 1) [4,14,15,16,17,18,19,20,21,22,23,24,25,26,27,28,29,30,31,32,33,34,35]. The majority of the papers included (21 studies) were published in the last decade, with 10 papers published from 2018 to 2020 [14,16,17,19,21,22,23,26,28,32] (Table 1).

### 3.2. Study Characteristics

The summary of selected studies is shown in Table 1. Regarding the type of study, five were RCTs [20,23,27,29,31] and 18 were non-randomized studies, of which 11 papers were interrupted time series studies (ITSSs) [4,14,15,17,18,19,22,25,26,28,32], two were controlled before and after studies (CBAs), and five were before and after comparison studies (BAs) [16,30,33,34,35]. Regarding the type of study design, 12 studies were single-arm trials [4,15,16,17,18,19,25,26,28,30,33,34], 10 were double-arm trials [14,20,21,22,23,24,29,31,32,33], and one was a multi-arm trial [27]. Eight papers had a sample size greater than 50 participants [4,17,19,21,22,27,28,32]. Regarding study length, only 13 studies lasted more than six months [14,15,16,17,18,19,21,22,24,25,27,28,30], with a follow-up evaluation that ranged from one week to 12 months.

### 3.3. Subjects Involved

An overall 1106 children with ASDs were included and evaluated, of which 532 were recruited for oral hygiene skills assessment and 574 for cooperation during dental treatment assessments after a visual pedagogy intervention. The patients’ ages ranged from 3 to 23 years, with an overall minimum average age of 4.50 years and maximum of 12.28 years. Thirteen studies reported the intellectual disability level of the children involved [4,17,20,21,22,24,26,27,28,29,31,32,33].

### 3.4. Visual Pedagogy Tools and Protocol

The Pictured Exchanged Communication System was used in eight studies [14,15,16,23,25,26,29,35], other kinds of images were used in 10 studies [4,17,18,19,21,24,28,30,31,34], and video and/or video plus images were used in five studies [20,22,27,32,33]. The intervention protocol foresaw the administration of visual tools on a daily basis in 11 studies [14,15,16,17,18,19,20,21,22,30,31], on weekdays in one study [28], on a weekly basis in six studies [4,26,29,32,33,34], once in five studies [23,24,25,27,35], and on a weekly followed by a daily basis in one study [4].

### 3.5. Oral Hygiene Outcome

Oral hygiene skills improvement was assessed using two clinical outcomes: the Gingival Index (GI) and the Plaque Index (PI). Tooth brushing performance, as the number of subsequent steps acquired in a tooth brushing session, was also used (Table 2).

Plaque index (PI) was measured in eight studies, six of which used the Silness and Löe Index [14,15,16,17,18,22], one study used the Podshadley and Haley Index [20], and one study used the Simplified Debris Index [21]. In four studies, the Gingival index (GI) was evaluated, three of which used the Löe and Silness Index [14,17,22], and one study used the Modified Gingival Index [21]. Tooth brushing performance was evaluated in two studies, where the tooth brushing session was split in five [18] and 13 [20] steps [19,21]. All studies included reported an improvement in tooth brushing performance and/or PI and GI indexes of ASD children after intervention with visual tools, and this was statistically significant (*p* < 0.05) in all [14,16,17,19,20,21,22] except two studies [15,18].

### 3.6. Dental Care Outcomes

Visual pedagogy efficacy was evaluated during dental examination in 13 studies [4,23,24,25,26,27,28,29,30,31,32,33,35] and at orthodontic check-up in one study [34]. In addition, the following dental procedures were evaluated: professional teeth cleaning [4,23,24,25,29,30,35], topical fluoride applications [24,31,35], sealants application [4,25], radiographic examination [24], restorative procedures [4,25], and surgical procedure [25]. The following variables were used to measure the ability of children with ASDs to perform a dental procedure: number of patients who were able to complete a dental procedure [4,24,25,31], number of attempts for each skill acquisition [23], number of visits to complete a dental treatment [23,24], time (minutes) spent to perform a skill [29,34], and finally, number of steps completed within a dental procedure, considering a variable number of steps from 6 to 13 for each procedure, such as a dental visit or professional oral hygiene, quite different from paper to paper [26,29,32,33,34]. The steps common to all studies included entering the dentist’s room, sitting in the dental chair, opening the mouth, and accepting the mouth mirror inside the oral cavity (Table 3).

The cooperation of children with ASDs during dental treatment was measured by the means of scores assigned according to the Frankl Behavior Scale in four studies [28,30,32,33], the Likert Anxiety Scale in one study [29], and the Venham Behavior Scale in one study [27].

All studies included reported an improved cooperation level of children with ASDs during dental procedures after intervention with visual pedagogy, and this was statistically significant in 10 studies (*p* < 0.05) [4,23,26,27,28,30,31,32,33,35] (Table 3).

### 3.7. Risk of Bias Assessment

Regarding the five RCTs (Figure 2), four were judged at a moderate risk of bias [9,23,27,31] and one at a high risk of bias [20]; among the 18 non-randomized studies (Table 4), one was judged at a low risk of bias [22], 16 were at a moderate risk of bias [4,14,15,16,17,18,19,21,24,25,26,28,30,32,33,35], and one was at a serious risk of bias [34].

Bias arising from the measurements of the outcomes significantly affected the quality rating of both the RCTs and the NRSIs. The randomization process aroused some concerns in more than 75% of RCTs (Figure 2), while confounding variables were not properly controlled in almost all NRSIs (Table 4).

### 3.8. Meta-Analysis

Data from four studies [14,15,17,22] were aggregated for meta-analysis, and a subgroup analysis by the type of tool was performed to assess the use of PECS and other types of visual tools (non-PECS) on Plaque Index results after six months (Figure 3).

The effect size was calculated within each group and across all studies using an inverse-variance model. Sub-group heterogeneity was moderate both in PECS (I^2^ = 28.99%) and non-PECS (I^2^ = 26.00%), while overall heterogeneity was high (I^2^ = 81.32%). Both PECS and non-PECS aids were effective in PI improvements, but no differences were found between the two sub-groups (*p* = 0.34).

## 4. Discussion

Visual pedagogy has been proposed as an effective approach to allow children with ASDs to become familiar with a dental environment, help them cope during outpatient procedures, and learn oral hygiene skills to maintain good oral health status. This method is widely used at home and at school for daily life activities and educational purposes; it is based on the visual receptivity of pictures, photos, and videos, which enable communication in non-verbal and/or non-fluent patients, the learning of new activities or social cues, and a reduction of anxiety when dealing with unfamiliar situations [10].

The systematic review was designed and carried out to assess whether visual pedagogy is an effective tool for oral hygiene and outpatient dental care in children with ASDs.

Oral hygiene studies showed that visual pedagogy is effective in improving and maintaining good oral health in patients with ASDs, as revealed by improvement of PI and GI in all of the studies performing this evaluation. Almost all studies investigating behavior during dental care showed an increased cooperation of children. Overall, visual pedagogy is effective in improving oral hygiene/tooth brushing skills and cooperation levels in dental settings.

This method of dental management has been only recently investigated, as revealed by the small sample of eligible articles selected for this systematic review, mostly published in the last decade. Despite the few papers included in this systematic review, 1142 children with ASDs were evaluated, representing a good sample size to provide some considerations on this topic.

The risk of bias was present in all kinds of studies due to poor stratification and lack of homogeneous samples. The majority did not differentiate the ASD level, verbal fluency, and/or previous use of visual tools. Drop-out rates might be the consequence of involving patients who in any case would not be able to adequately comply to visual pedagogy, rather than a failure of the treatment itself. Patient selection and outcome measurements should be performed based on factors that can predict the patient’s assignment to and/or performance in using visual tools to better outlying limits and indications of visual pedagogy in dental settings. A behavioral approach to dental care with children might be affected by an inner and unavoidable inter-operator variability that is difficult to reduce, even when treatment procedures are well-standardized [40].

Many NRSIs were single-arm studies, lacking a control group. It is important to perform such intervention studies in at least a double-arm design to avoid drawing inconsistent conclusions. Intervention studies on children with ASDs often struggle to have adequate blinding to overcome measurements bias. The reason is that these patients often require specialized dental teams working in environments dedicated to special needs children, where it is not always possible to have adequate personal staff.

The high heterogeneity of treatment protocols in studies evaluating the behavior of children with ASDs during dental treatment has made it difficult to compare results among different studies and not possible to develop a meta-analysis. Standardized visual pedagogy protocols should be planned by establishing a narrow range of both the frequency and types of visual tools used, with adequate validation of patients’ and parents’ cooperation by, for example, means of questionnaires. The majority of the studies evaluated cooperation during non-invasive and/or minimally invasive procedures; however, visual pedagogy needs to be evaluated also in invasive and/or more complex treatments, since its efficacy during oral check-ups has already been validated.

The meta-analysis performed on the four studies confirmed that visual supports are effective. The meta-analysis also addressed any differences between PECS and non-PECS visual supports: PECS revealed a slightly better performance, but no consistent conclusion can be drawn.

## 5. Conclusions

The wide heterogenicity of studies included in this systematic review does not allow for the conclusion of clear evidence on the effectiveness of visual pedagogy in dental settings. Nevertheless, its use improved both oral hygiene skills and cooperation during dental care in children with ASDs, even if it is not possible to clarify which visual tool is more effective.

## Figures and Tables

**Figure 1 ijerph-18-00789-f001:**
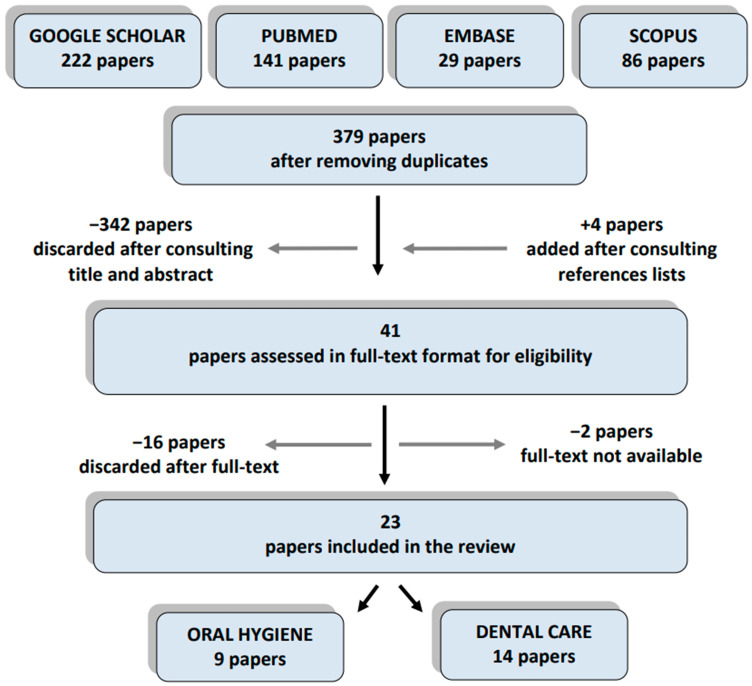
Flow chart of the search.

**Figure 2 ijerph-18-00789-f002:**
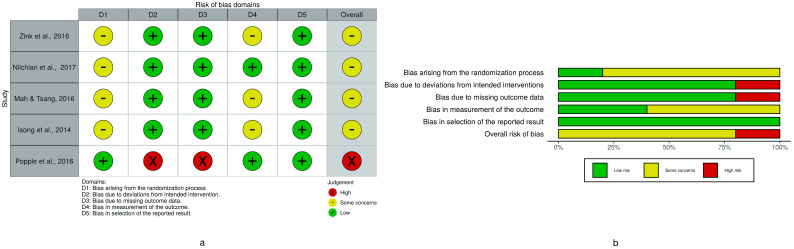
Risk of bias assessment of RCTs using the ROB-2 tool. (**a**) Traffic light plot of RCT bias assessment. (**b**) Weighted summary plot of the overall type of bias encountered in RCTs.

**Figure 3 ijerph-18-00789-f003:**
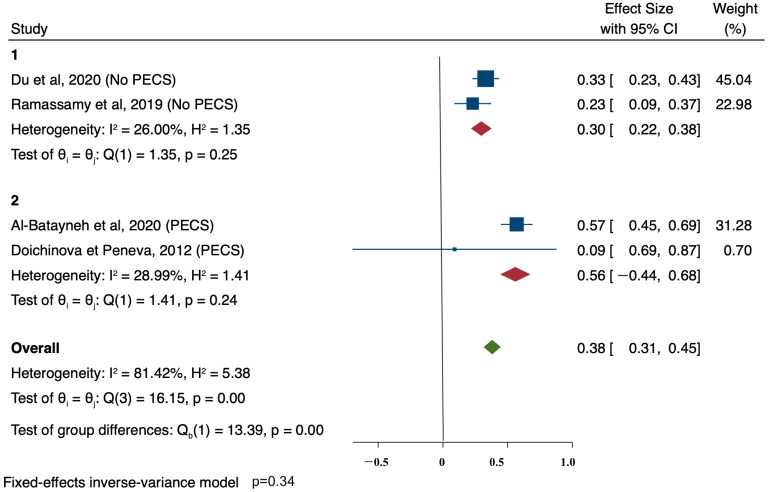
Meta—analysis of Plaque Index outcomes and subgroup analysis by the type of tool (PECS vs. other type of visual non-PECS).

**Table 1 ijerph-18-00789-t001:** General characteristics of the studies included regarding the use of visual tools in ASD children’s oral hygiene and dental care.

Authors	Sources	Location	Database	Type of Study	Aim	Risk Of Bias Assessment
Du et al. [17]	Int. J. Paediatr. Dent. 2021, 31, 89–105	Hong Kong (China)	PM, E	ITSS	Oral Hygiene	Moderate
Al-Batayneh et al. [14]	Eur. Arch. Paediatr. Dent. 2020, 21, 277–283	Irbid (Jordan)	PM, E, S	ITSS	Oral Hygiene	Moderate
Zhou et al. [21]	Autism. Res. 2020, 13, 666–674	Hong Kong (China)	PM, E, GS	CBA	Oral Hygiene	Moderate
Doichinova et al. [16]	Biotechnol. Biotechnol. Equip. 2019,33, 748–755	Sofia (Bulgaria)	GS	BA	Oral Hygiene	Moderate
Orellana et al. [32]	Med. Oral Patol. Oral Cir. Bucal. 2019, 24, 37–46	BIO-BIO region (Chile)	PM, GS, S	ITSS	Dental Care	Moderate
Lopez-Cazaux et al. [19]	Eur. Arch. Paediatr. Dent. 2019, 20, 277–284	Nantes (France)	PM, GS, S	ITSS	Oral Hygiene	Moderate
Lefer et al. [28]	Eur. Arch. Paediatr. Dent. 2019, 20, 113–121	Nantes (France)	PM, S	ITSS	Dental Care	Moderate
Ramassany et al. [22]	Spec. Care Dentist. 2019, 39, 551–556	Puducherry (India)	PM, E, GS, S	ITSS	Oral Hygiene	Low
Hidayatullah et al. [26]	Dent. J. 2018, 51, 71–75	Bandung (Indonesia)	GS	ITSS	Dental Care	Moderate
Zink et al. [23]	Pediatr. Dent. 2018, 40, 18–22	Sao Paolo (Brazil)	PM, E, GS, S	RCT	Dental Care	Moderate
Murshid. [30]	Saudi. Med. J. 2017, 38, 533–540	Riyadh (Saudi Arabia)	PM	BA	Dental Care	Moderate
Nilchian et al. [31]	J. Autism. Dev. Disord. 2017, 47, 858–864	Isfahan (Iran)	PM, E, GS	RCT	Dental Care	Moderate
Popple et al. [20]	J. Autism. Dev. Disord. 2016, 46, 2791–2796	New Haven (USA)	PM, GS	RCT	Oral Hygiene	Moderate
Zink et al. [35]	Spec. Care Dentist. 2016, 36, 254–259	Sao Paolo (Brazil)	PM, S	BA	Dental Care	Moderate
Mah & Tsang [29]	J. Clin. Pediatr. Dent. 2016, 40, 393–399	Vancouver (Canada)	PM, E, GS, S	RCT	Dental Care	Moderate
Cagetti et al. [4]	Med. Oral Patol. Oral Cir. Bucal. 2015, 20, 598–604	Milan (Italy)	PM, GS, S	ITSS	Dental Care	Moderate
Isong etal. [27]	Clin. Pediatr. 2014, 53, 230–237	Boston (USA)	PM, GS	RCT	Dental Care	Moderate
Schindel etal. [34]	J. Clin. Orthod. 2014, 48, 285–291	Commack (USA)	PM, E	BA	Dental Care	Serious
Bossù et al. [25]	Senses Sci. 2014, 1, 107–112	Rome (Italy)	GS	ITSS	Dental Care	Moderate
Orellana et al. [33]	J. Autism. Dev. Disord. 2014, 44, 776–785	Valencia (Spain)	PM, GS	BA	Dental Care	Moderate
Doichinova & Peneva [15]	Prob. Dent. Med. 2012, 38, 12–18	Sofia (Bulgaria)	GS	ITSS	Oral Hygiene	Moderate
Pilebro & Bäckman [13]	Int. J. Paediatr. Dent. 2005, 15, 1–9	Umea (Sweden)	PM, E, GS, S	ITSS	Oral Hygiene	Moderate
Bäckman & Pilebro [24]	J. Dent. Child. 1999, 66, 325–331	Umea (Sweden)	PM, S	CBA	Dental Care	Moderate

BA: Before and after comparison study; CBA: controlled before and after study; ITSS: interrupted time series study; RCT: randomized controlled trial; PM: PubMed; S: Scopus; E: Embased; GS: Google Scholar.

**Table 2 ijerph-18-00789-t002:** Main characteristics of the included studies regarding the effectiveness of visual pedagogy in improving ASD children’s skills in oral hygiene.

Author (Year)	N-Subjects M/F Age-Range	Intellectual Disability/Verbal Fluency	Study Length	Type of Tool	Visual Pedagogy Protocol	Adjunctive Tool	Study Design/Groups (Outcome)	Results Mean (SD)		Findings
**PECS**										
Al-Batayneh et al. (2020) [14]	37	Fluent, non-fluent and non-verbal	Six mo	PECS (paper)	Provided daily by parents/caregivers	-	Two groups:	G1	G2	PI and GI showed a statistically significant improvement at three-month and six-month evaluations in both groups (*p* < 0.01). No comparison between groups was performed
	M/F						G1: 4–10 yy (*n* = 24)	Plaque Index (PI)		
	4–16 yy						G2: 10–16 yy (*n* = 13)	Baseline 1.88 (0.36)	Baseline 2.17 (0.26)	
								Three mo 1.47 (0.30)	Three mo 1.47 (0.26)	
							GI and PI	Six mo 1.27 (0.34)	Six mo 1.38 (0.24)	
								Gingival Index (GI)		
								Baseline 1.12 (0.22)	Baseline 1.26 (0.23)	
								Three mo 0.89 (0.19)	Three mo 0.97 (0.28)	
								Six mo 0.85 (0.17)	Six mo 0.95 (0.27)	
Doichinova et al. (2019) [16]	30	Non-verbal and non-fluent	12 mo	PECS (paper)	Provided daily by parents/caregivers	Behavioral management; TSD	One group (*n* = 30)	Plaque Index (PI)		PI showed a statistically significant improvement at three-, six-, and 12-month evaluations (*p* < 0.05)
	-							Baseline 2.29		
	6–11 yy						PI	Three mo 1.95 (0.36)		
								Six mo 1.88 (0.35)		
								12 mo 1.79 (0.36)		
Doichinova & Peneva, (2012) [15]	30	Moderately severe	12 mo	PECS (paper)	Provided daily by parents and for 15 min every two weeks by dental specialist	Behavioral management; TSD	One group (*n* = 30)	Plaque Index (PI)		PI improved but did not reach a statistical significance (*p* > 0.05)
	M/F							Baseline 2.49 (0.55)		
	4–11 yy						PI	Three mo 2.40 (0.15)		
								Six mo 2.42 (0.21)		
								12 mo 2.34 (0.21)		
**Different Kind of Images**										
Du, (2020) [17]	122	From mild to severe	Six mo	Photos (paper)	Provided daily by parents/caregiver	-	One group (*n* = 122)	Plaque Index (PI)		PI and GI showed a statistically significant improvement at three- and six-month evaluations (*p* < 0.01)
	M/F							Baseline 1.00 (0.32)		
	2.5–7 yy						GI and PI	Three mo 0.67 (0.27)		
								Six mo 0.63 (0.25)		
								Gingival Index (GI)		
								Baseline 0.91 (0.26)		
								Three mo 0.58 (0.26)		
								Six mo 0.60 (0.26)		
Zhou, (2020) [21]	169	From mild to severe	Six mo	Social Story (paper)	Provided daily by parents/caregivers	-	Two groups:	G1	G2	Tooth brushing performance, DI-S, and MGI showed a statistically significant improvement at the six-month evaluation in both groups (*p* < 0.01). Children with ASDs showed better oral hygiene status (*p* = 0.01) and gingival status (*p* < 0.01) than their peers with other disabilities. No significant difference in the tooth brushing performance between groups was found.
	-						G1: ASD (*n* = 84)	Plaque Index (DI-S)		
	<6 yy						G2: other disability (*n* = 85)	Baseline 1.63 (0.82)	Baseline 1.64 (0.77)	
								Six mo 0.68 (0.42)	Six mo 0.85 (0.44)	
								Gingival Index (MGI)		
							Tooth brushing performance, DI-S, and MG	Baseline 1.02 (0.64)	Baseline 1.17 (0.56)	
								Six mo 0.43 (0.42)	Six mo 0.69 (0.50)	
							Tooth brushing performance, DI-S, and MGI	Tooth brushing performance (steps achieved)		
								Baseline 6.69 (3.23)	Baseline 6.62 (2.69)	
								Six mo 8.30 (3.36)	Six mo 8.07 (3.41)	
Lopez Cazaux et al. (2019) [19]	52	-	Eight mo	Pictograms and photos (digital)	Provided daily by parents/caregivers and weekly by dentist	-	One group (*n* = 52)	Tooth brushing performance *		Tooth brushing performance showed a statistically significant improvement at the eight-month evaluation at each step (*p* < 0.01)
	M/F							Put toothpaste on the brush		
	3–19 yy						Tooth brushing performance	Baseline 4.2 (0.8)		
								Four mo 4.5 (0.7)		
								Eight mo 4.8 (0.5)		
								Brush occlusal surface		
								Baseline 3.1 (1.0)		
								Four mo 3.9 (1.0)		
								Eight mo 4.2 (0.7)		
								Brush buccal surface		
								Baseline 2.6 (1.2)		
								Four mo 3.5 (0.9)		
								Eight mo 3.8 (0.9)		
								Brush lingual surface		
								Baseline 2.1 (1.0)		
								Four mo 3.4 (1.0)		
								Eight mo 3.8 (0.9)		
								Spit and store the brush		
								Baseline 3.9 (0.9)		
								Four mo 4.2 (0.8)		
								Eight mo 4.3 (0.8)		
Pilebro & Bäckman, (2005) [18]	14	Fluent and non-fluent	18 mo	Photo (paper)	Provided daily by parents	-	One group (*n* = 14)	Plaque Index (PI)		PI showed improvement at eight- and 12-month evaluations (no statistical analysis available)
	M							Baseline 2.57		
	5–13 yy						PI	Eight mo 1.64		
								12 mo 1.92		
**Videos or Videos Plus Images**										
Ramassamy et al. (2019) [22]	67	Moderate	Six mo	Pictures (paper) and video	Provided daily by parents or teacher	-	Two groups:	G1	G2	Children in G2 demonstrated better oral hygiene. PI and GI were statistically significantly different at two months (*p* = 0.04 and *p* = 0.01), three months (*p* = 0.01 and *p* = 0.01), and six months (*p* = 0.01 for both) between groups.
	M/F						G1: visual pedagogy (*n* = 32)	Plaque Index (PI)		
	7–15 yy						G2: visual pedagogy + yoga therapy (*n* = 35)	Baseline 1.78 (0.14)	Baseline 1.75 (0.25)	
								Three mo 1.55 (0.21)	Three mo 1.22 (0.39)	
								Six mo 1.35 (0.35)	Six mo 0.96 (0.34)	
							PI and GI	Gingival Index (GI)		
								Baseline 1.76 (0.14)	Baseline 1.72 (0.22)	
								Three mo 1.59 (0.17)	Three mo 1.36 (0.36)	
								Six mo 1.49 (0.18)	Six mo 1.09 (0.27)	
Popple et al. (2016) [20]	18	Moderate	Six wk	Video	Provided daily by parents	-	Two groups:	G1	G2	PI-PH showed a statistically significant improvement at three-week and six-week evaluations in both groups (*p* < 0.01); statistically significant differences between G1 and G2 at the six-week evaluation were found (d = 1.02)
	M/F						G1: intervention video (*n* = 9)	Plaque Index (PI-PH)		
	5–14 yy						G2: control video (*n* = 9)	Baseline 1.78 (0.62)	Baseline 1.75 (0.83)	
								Three wk 0.92 (0.65)	Three wk 1.45 (0.91)	
							PI-PH	Six wk 0.38 (0.43)	Six wk 1.20 (1.05)	

yy: Years; mo: months; wk: weeks; M: male; F: female; PECS: Picture Exchange Communication System; TSD: tell-show-do; PI: Plaque Index (Löe and Silness Index); GI: Gingival Index (Silness and Löe Index); DI-S: Simplified Debris Index; MGI: Modified Gingival Index; PI-PH: Plaque Index (Podshadley and Haley); * results rounded to one decimal.

**Table 3 ijerph-18-00789-t003:** Main characteristics of the included studies regarding ASD children’s behavior during dental care procedures.

Author (Year)	N-Sunjects M/F Age-Range	Intellectual Disability/Verbal Fluency	Study Length	Type of Tool	Visual Pedagogy Protocol	Adjunctive Tool	Study Design/Groups (Outcome)	Dental Visit Results Mean (SD) or Counts	Findings
**PECS**
Zink et al. (2018) [23]	40	-	-	Images (digital) or PECS (paper)	Provided once by dentist	BM	Two Groups:		G1	G2	The mean number of attempts for acquiring each step (*p* < 0.01) and number of visits to fully cooperate (*p* < 0.01) were significantly lower in G1 compared to G2
M/F	G1: app on iPad^®^ (*n* = 20)	Mean number of attempts (*n*) *
9–15 yy	G2: PECS (*n* = 20)	Step One	1.1 (0.3)	1.5 (0.8)
		Step Two	1.1 (0.5)	1.6 (0.6)
	Professional teeth-cleaning, divided in seven steps, reported as the number of attempts for acquiring each step and number of visits to fully cooperate	Step Three	1.5 (0.5)	1.6 (0.7)
	Step Four	1.0 (0.2)	1.7 (1.1)
	Step Five	1.2 (0.4)	1.6 (0.9)
Step Six	1.2 (0.6)	1.8 (0.9)
Step Seven	2.1 (0.9)	2.6 (1.7)
Number of visits (*n*) *
	3.0 (1.0)	4.3 (1.2)
Hidayatullah et al. (2018) [26]	13	From mild to moderate	One mo	PECS (paper)	Provided weekly by teacher and by dentist	BM	One group (*n* = 13)	Mean number of steps completed (*n*) ^†^	Patients were able to perform more steps within a dental visit at one-, two-, three-, and four-week evaluations (*p* < 0.01)
M/F	One wk	1.9 (1.3)	
5–18 yy	Dental visit divided into 10 steps	Two wk	3.5 (0.9)	
	Three wk	4.5 (1.7)	
Four wk	5.6 (1.9)	
Zink et al. (2016) [35]	26	-	-	PECS (paper)	Provided once by the dentist	BM	Two groups		G1	G2	The mean number of attempts required for steps two, four, five, and six were significantly lower (*p* < 0.05) in G1 compared to G2
M/F	G1: no dental experience (*n* = 13)	Mean number of attempts (*n*) *
5–19 yy
M/F	G2: previous dental experience (*n* = 13)	Step One	1.8 (1.0)	1 2.8 (1.6)
Step Two	1.5 (1.5)	2.5 (1.9)
	Step Three	2.8 (1.6)	4.4 (2.5)
Dental visit divided into six steps, and professional teeth-cleaning (including fluoride therapy), considered as step seven, reported as the number of attempts for acquiring each step	Step Four	2.0 (1.9)	3.4 (2.1)
Step Five	2.1 (1.7)	4.6 (2.1)
Step Six	2.5 (2.1)	4.4 (1.7)
Step Seven	3.8 (3.3)	4.6 (2.4)

Mah and Tsang, (2016) [29]	14	Mild	Three wk	PECS (paper)	Provided weekly by dentist/hygienist	TSD	Two groups		G1	G2	The mean number of steps completed increased, completion time decreased, and anxiety decreased more in G1 compared to G2 at one-, two-, and three- week evaluations (no statistical analysis available)
M	G1: test group (*n* = 7)	Mean number of steps completed (*n*)
4–8 yy	G2: control group (*n* = 7)	Baseline	8.91 (2.04)	7.95 (2.04)
		One wk	10.12 (1.81)	8.31 (1.81)
	Dental visit, divided into seven steps, and professional teeth-cleaning, divided in the following five steps, reported as the number of steps completed at each visit and time to perform them	Two wk	10.54 (1.68)	9.17 (1.68)
	Three wk	11.48 (1.28)	10.09 (1.28)
	Completion time per step (min)
	Baseline	1.41 (0.47)	7.95 (2.04)
	One wk	1.04 (0.35)	8.31 (1.81)
	Two wk	1.00 (0.43)	9.17 (1.68)
	Three wk	0.98 (0.45)	10.09 (1.28)
	Anxiety Likert Scale Score
	Baseline	1.55 (0.48)	2.48 (0.48)
	One wk	1.62 (0.45)	2.14 (0.45)
	Two wk	1.77 (0.57)	2.15 (0.57)
	Three wk	1.65 (0.54)	2.08 (0.54)
Bossù et al. (2014) [25]	34	-	Three yy	PECS (paper)	Provided once by the dentist	TSD, desensitization	One group (*n* = 34)	Cooperative patients (*n*)	The majority of children were cooperative during dental procedures (no statistical analysis available)
M/F		Tooth extraction 2/2
6–12 yy	Number of cooperative patients during preventive, restorative, and surgical procedures	Tooth Filling 8/10
	Oral hygiene 30/34
	Dental sealant 18/28
**Different Kind of Images**
Lefer et al. (2019) [28]	52	From mild to severe	Eight mo	Photos (digital)	Provided at weekdays by teacher	-	One group (*n* = 52)	Global Skill acquisition Score *	Both scores significantly improved at two-, four-, six-, and eight-month evaluations (*p* < 0.01) compared to baseline
M/F		Baseline	2.3 (0.6)	
3–19 yy	Dental visit, divided into six steps, reported as a score from 1 to 3 (1 = not acquired; 2 = emerging; 3 = acquired), and behavior assessment (Frankl Behavior Score)	Two mo	2.5 (0.5)	
	Four mo	2.7 (0.5)	
	Six mo	2.7 (0.4)	
	Eight mo	2.8 (0.6)	
	Frankl Behavior Score *
	Baseline	2.7 (0.8)	
	Two mo	3.2 (0.9)	
	Four mo	3.3 (1.0)	
	Six mo	3.4 (0.8)	
	Eight mo	3.4 (0.9)	
Murshid E. Z. (2017) [30]	40	-	Six mo	Dental book (Paper)	Provided daily by parents	-	One Group (*n* = 40)	Cooperative patients (%) *	Children’s behavior significantly improved at a four-month evaluation (*p* < 0.01)
M/F		One wk		
5–9 yy	Dental visit, professional teeth-cleaning, and fluoride therapy, reported as the number of patients for each behavior score (Frankl Behavior Score)		Definitely positive	0
		Positive	47.5
		Negative	32.5
		Definitely negative	20.0
	Four mo		
		Definitely positive	0
		Positive	80.0
		Negative	12.5
		Definitely negative	7.5
Nilchian et al. (2017) [31]	40	From mild to moderate	Two mo	Coloring pictures (Paper)	Provided twice a day by a trained teacher	-	Two Groups:		G1	G2	Number of cooperative patients increased significantly at two-, four-, six-, and eight-week evaluations in both groups (*p* < 0.01) and at eight weeks during fluoride therapy only in G1 (*p* < 0.01). No further inter-group differences were found
M/F	G1: case group (*n* = 20)	Cooperative patients (%)
6–12 yy	G2: control group (*n* = 20)	Dental visit
		Baseline	15	15
	Number of cooperative patients during dental visit and fluoride therapy	Two wk	30	15
	Four wk	40	25
	Six wk	50	55
	Eight wk	70	65
	Fluoride therapy
	Baseline	0	0
	Two wk	0	0
	Four wk	0	0
	Six wk	5	0
	Eight wk	30	5
Cagetti et al. (2015) [4]	83	From mild to severe	One and a half mo	Coloring pictures (digital)	Provided twice a week by psychologist, then daily by parents	-	One group (*n* = 83)	Cooperative patients (*n*)	The majority of children were cooperative (no statistical analysis available). Cooperation was statistically significantly influenced by the child’s verbal fluency in all treatments (*p* ranging from 0.04 to 0.01) and by intellectual disability in restorative treatment (*p* = 0.04)
M/F	Number of cooperative patients during dental visit and preventive and restorative treatments	Oral examination 77/83
6–12 yy	Teeth-cleaning 77/77
	Sealant 70/77
	Restoration 41/44
Schindel et al. (2014) [34]	16	-	Two wk	Photos (Paper)	Provided twice a day by a trained teacher	TSD	One group (*n* = 20)	Mean number of steps completed (*n*) ^†^	The mean number of steps completed increased and time required per step decreased at a two-week evaluation in all children except one.
M/F		Baseline	6.37 (5.10)	
10–23 yy


	Orthodontic examination, divided into 13 steps, reported as the number of steps completed and time to perform them.	Two wk	11.12 (4.18)	
Mean completion time (min) ^†^
Baseline	18.78 (7.08)	
Two wk	11.54 (4.58)	
Bäckman and Pilebro, (1999) [24]	32	From mild to severe	18 mo	Dental book (Paper)	Provided once or more by parents	-	Two groups:		G1	G2	Children in G1 were more cooperative compared to G2 (no statistical analysis available)
M/F	G1: visual pedagogy (*n* = 16)	Number of cooperative patients (*n*)
3–6 yy	G2: control group (*n* = 16)	Dental visit	11	4
		Fluoride therapy	2	0
	Number of cooperative patients during dental visit, tooth-cleaning, and fluoride therapy	Teeth-cleaning	4	0
**Video or Video + Images**
Orellana et al. (2019) [32]	74	From moderate to severe	-	Video	Provided weekly by dentist	TSD, BM, PM	2 groups:		G1	G2	The mean number of steps completed increased (*p* < 0.01) and the children’s behavior improved (*p* < 0.01) at a seven-week evaluation and was maintained after one month.
M/F	G1: 4–9 yy group (*n* = 52)	Mean number of steps completed (*n*) *
4–17 yy	G2: 10–17 yy group (*n* = 22)	Baseline	3.9 (2.7)	4.4 (2.6)
		One mo	9.4 (1.5)	9.5 (1.7)
	Dental visit, divided in 10 steps, reported as the number of steps completed and behavior assessment (Frankl Behavior Score)	Frankl Behavior Score *
	Baseline	1.9 (0.8)	2.0 (0.8)
	One mo	3.3 (0.8)	3.5 (0.7)
Isong et al. (2014) [27]	80	From mild to moderate	Six mo	Video	Once or more at the home with parents	-	Four groups:		G1	G2	G3	G4	Anxiety and behavior scores statistically significantly decreased at a six-month evaluation in G3 and G4 (*p* < 0.05).
M/F	G1: control group (*n* = 20)	Venham Behavior Scale *
7–17 yy	G2: video peer model (*n* = 20)	Baseline	2.2 (1.9)	2.7 (1.8)	2.5 (1.6)	2.9 (1.5)
	G3: video googles (*n* = 20)	Six mo	2.3 (1.6)	2.9 (2.0)	1.7 (1.9)	2.1 (1.6)
	G4: video model + goggles (*n* = 20)	Venham Anxiety Scale *
		Baseline	2.4 (1.8)	2.6 (1.8)	2.6 (1.3)	2.9 (1.3)
	Dental visit, reported as a behavior and anxiety assessment (Venham scores)	Six mo	2.3 (1.6)	2.6 (1.9)	1.7 (1.8)	2.1 (1.6)
Orellana et al. (2014) [33]	38	From mild to severe	Four wk	Images (paper) and video	Provided twice a week by dentist	TSD, desensitization, and modelling	1 group (*n* = 38)	Mean number of steps completed (*n*) ^†^	The mean number of steps completed statistically significantly increased (*p* < 0.01) and children’s behavior improved (*p* < 0.01) at a four-week evaluation. Improvements were observed in high-functioning children as well as children with mild and severe disability.
M/F		Baseline	3.03 (2.22)	
4–10 yy	Dental visit, divided into 10 steps, reported as the number of steps completed and behavior assessment (Frankl Behavior Score)	Four wk	9.03 (2.05)	
	Frankl Behavior Score *
	Baseline	1.95 (0.77)	
	Four wk	3.24 (0.88)	

yy: Years; mo: months; wk: weeks; M: male; F: female; PECS: Picture Exchange Communication System; TSD: tell-show-do; BM: behavioral management; PM: peer modelling; min: minutes; * results rounded to one decimal; † mean and standard deviation calculated by reviewers when raw data available.

**Table 4 ijerph-18-00789-t004:** Risk of bias assessment of non-randomized studies of intervention (NRSI) using the ROBINS-I tool.

Study	Confounding	Selection of Participants	Classification of Interventions	Deviations from Intervention	Missing Data	Measurements of Outcome	Selection of the Reported Results	Overall
	**ORAL HYGIENE**							
Du et al., 2020 [17]	Serious	Low	Low	Low	Moderate	Serious	Low	Moderate
Al-Batayneh et al., 2020 [14]	Critical	Low	Low	Serious	Moderate	Serious	Low	Moderate
Zhou et al., 2020 [21]	Moderate	Low	Low	Moderate	Low	Serious	Low	Moderate
Doichinova et al., 2019 [16]	Serious	Low	Moderate	Serious	Low	Serious	Low	Moderate
Lopez Cazaux et al., 2019 [19]	Critical	Low	Low	Moderate	Low	Serious	Low	Moderate
Ramassamy et al., 2019 [22]	Moderate	Low	Low	Low	Low	Serious	Low	Low
Doichinova & Peneva, 2012 [15]	Critical	Low	Moderate	Moderate	Low	Serious	Low	Moderate
Pilebro & Bäckman, 2005 [18]	Serious	Low	Low	Low	Low	Serious	Low	Moderate
	**DENTAL CARE**							
Orellana et al., 2019 [32]	Moderate	Low	Low	Low	Low	Serious	Moderate	Moderate
Lefer et al., 2019 [28]	Serious	Low	Low	Low	Low	Serious	Low	Moderate
Hidayatullah et al., 2018 [26]	Serious	Low	Low	Low	Low	Serious	Low	Moderate
Murshid, 2017 [30]	Critical	Low	Low	Serious	Low	Low	Low	Moderate
Zink et al., 2016 [23]	Critical	Low	Low	Low	Low	Serious	Low	Moderate
Cagetti et al., 2015 [4]	Serious	Low	Low	Moderate	Low	Serious	Low	Moderate
Schindel et al., 2014 [34]	Critical	Serious	Serious	Low	Moderate	Serious	Critical	Serious
Bossù et al., 2014 [25]	Serious	Serious	Low	Moderate	Low	Serious	Low	Moderate
Orellana et al., 2014 [33]	Serious	Low	Low	Low	Moderate	Serious	Low	Moderate
Bäckman & Pilebro, 1999 [24]	Moderate	Low	Moderate	Serious	Low	Serious	Low	Moderate

Red color = Critical risk of bias; Orange color = Serious risk of bias; Yellow color = Moderate risk of bias; Green color = Low risk of bias.

## Data Availability

Not applicable.

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
