# Peer review of "Is Visual Pedagogy Effective in Improving Cooperation towards Oral Hygiene and Dental Care in Children with Autism Spectrum Disorder? A Systematic Review and Meta-Analysis"

_ijerph, 2021, doi:10.3390/ijerph18020789_

Round 1
Reviewer 1 Report
I had the opportunity of revising this interesting systematic review regarding visual pedagogy effectiveness in improving oral hygiene in children with Autism spectrum disorder.
The review is well written and well performed and I think it could be suitable for publication after some minor changes:
-In the methods section there were "no limits" in the publication period but please indicate the last month you searched the databases.
-In the methods section I suggest to indicate a primary outcome (probably PI?).
-Figure 2 and Figure 4 are not fully visible in the present page format.
Best Regards
Author Response
Please, see the attached file

Reviewer 2 Report
This represents a great work from the authors. However, the work should be focused more so the reader can get out with a clear conclusion after reading this research. In the Abstract, there was 'no conclusive evidence' and then 'the visual pedagogy improved both,' Please include conclusions based on the study's results that are also not contradictory.
Please clearly define the visual pedagogy and be consistent with writing them, with first letters small or capital. Is VP similar to different than PECS, images, videos, etc.? In line 76, the authors stated that they compared between them. Please clarify.
There is a need to define what outcomes are measured and compared. Is dental attitude similar to children's cooperation during dental treatment? I may suggest that authors look at oral hygiene only and provide us with a conclusive conclusion on whether the VP improved it or not. They can study other outcomes in another paper.
Author Response
Please, see the attached file

Reviewer 3 Report
This is one of the better manuscripts I have reviewed. There are some minor problems with the English spellings (British versus American). The tables did format correctly, which may have been a problem with the journal's software. Please see the attached file for specific comments.

Author Response
Please, see the attached file

Round 2
Reviewer 2 Report
The authors have addressed my comments adequately. I have no other concerns.